# Translation Potential and Challenges of In Vitro and Murine Models in Cancer Clinic

**DOI:** 10.3390/cells11233868

**Published:** 2022-11-30

**Authors:** Yuan Long, Bin Xie, Hong C. Shen, Danyi Wen

**Affiliations:** 1Shanghai LIDE Biotech Co., Ltd., Shanghai 201203, China; 2China Innovation Center of Roche, Roche R & D Center, Shanghai 201203, China

**Keywords:** cancer models, PDX, CR cells, organoid, MiniPDX

## Abstract

As one of the leading causes of death from disease, cancer continues to pose a serious threat to human health globally. Despite the development of novel therapeutic regimens and drugs, the long-term survival of cancer patients is still very low, especially for those whose diagnosis is not caught early enough. Meanwhile, our understanding of tumorigenesis is still limited. Suitable research models are essential tools for exploring cancer mechanisms and treatments. Herein we review and compare several widely used in vitro and in vivo murine cancer models, including syngeneic tumor models, genetically engineered mouse models (GEMM), cell line-derived xenografts (CDX), patient-derived xenografts (PDX), conditionally reprogrammed (CR) cells, organoids, and MiniPDX. We will summarize the methodology and feasibility of various models in terms of their advantages and limitations in the application prospects for drug discovery and development and precision medicine.

## 1. Introduction

In the past few decades, driven by breakthroughs in cancer research, the number of anti-cancer drugs has soared [1]. Consequently, cancer patients are living longer, yet mortality and morbidity in the vast majority of cancer patients are still devastating [2]. More recently, despite remarkable progress in the research of targeted anti-cancer drugs and immunotherapies, there are still challenges associated with drug efficacy, toxicity, and resistance. Thus, the urgency remains for more transformative treatments in order to address these unmet medical needs. Unfortunately, in contrast to drugs for the treatment of other diseases, the likelihood for a new phase one oncology clinical compound to reach market approval is significantly lower [3,4]. One reason for the low success rate is the translatability of preclinical cancer models to patients in clinic. Not only do the defects of the research model limit the development of new drugs, they also hinder the progress of precision medicine [5,6]. So far, genetic, epigenetic, and environmental factors have been frequently invoked to account for the development of tumors. At the same time, the pathogenesis of cancer is yet to be further elucidated. Our understanding of disease biology is still not always sufficient to effectively prevent or accurately predict tumor progression. As such, cancer models that can mimic tumor genesis and development are helpful for mechanistic research on cancer and could potentially enable better treatment options for cancer patients. In this review, we aim to discuss the features, utility, advantages, and disadvantages of various preclinical cancer models.

## 2. Syngeneic Tumor Models

Syngeneic tumor models are established by subcutaneously or orthotopically injecting mouse tumor cell lines expanded in vitro into immunocompetent mice. Syngeneic tumor models have been used in cancer research and drug development for more than 50 years across various cancer types, including lung, breast, bowel, liver, and malignant melanoma [7,8,9]. Established in immunocompetent mice, the syngeneic models are therefore suitable for evaluating drugs that affect immunity. Mosely et al. examined the responses of anti-CTLA-4 and anti-PD-L1 therapies in a panel of commonly used murine syngeneic tumor models, including 4T1, MC38, B16F10 AP-3, CT26, and LL/2. These syngeneic tumor models displayed variable responsiveness to immunotherapies, with significant differential gene expression panels identified, correlating to the tumor immune infiltrates. These results suggest there are two phenotypes of models, including “inflamed” and “non-inflamed”. Choosing a suitable model is crucial for the research of immunotherapies [10]. Since cell lines can be expanded rapidly and reproducibly in vitro, syngeneic mouse models have the advantages of being relatively fast to establish, low-cost, and reproducible. However, the use of cell lines also leads to a lack of genomic, epigenetic, and micro-environmental heterogeneity in this model, which cannot mimic the complex situations of patients well [11]. Another disadvantage is that only a few cell lines can be harnessed to establish syngeneic mouse models, thereby limiting the applications of these models.

## 3. Genetically Engineered Mouse Models (GEMM)

Genetically engineered mouse models are autochthonous cancer models driven by the expression of oncogenes or deletion of tumor suppressors using genetic engineering technology. Research on oncogenes and the development of genetic engineering technology contributed to the invention of genetically engineered mice. These models primarily use tissue-specific promoters to drive oncogene expression or tissue-specific recombinases to drive the loss of tumor suppressors [12,13,14,15,16]. The genetically engineered mouse model is a spontaneous model that can mimic the entire precancerous lesion and tumorigenesis processes. Therefore, this model has been applied for biomarker development [17,18], cancer mechanism research, and preclinical drug testing [19,20,21]. For example, the transgenic adenocarcinoma of the mouse prostate (TRAMP) is a model of prostate cancer hormonally regulated by androgens. This model develops progressive, multifocal, and heterogeneous prostate cancer, correlating with sexual maturity, which closely mimics the clinical disease [19]. At the same time, a genetically engineered mouse model is also suitable for studying immunotherapy and immune-related drugs. Genetically engineered mouse models have several advantages, such as being autochthonous and carrying an intact immune system and microenvironment. However, cancer models caused by alterations in a tumor-driver gene do not fully mimic the complexity of human cancer mechanisms. Meanwhile, the tumor formation in GEMM is variable and has longer latency periods than transplantable models, thus requiring large colonies of mice and extended experimental timelines. In addition, only a handful of genetically engineered mouse models are available, thus limiting the use of this technology in cancer research.

## 4. Cell Lines and Cell Line-Derived Xenograft (CDX)

Traditional cancer cell lines as tools used in vitro and in vivo have made significant contributions to cancer research and drug discovery and development for decades [22]. A large amount of the original cell lines were established by the National Cancer Institute (NCI) and the Hamon Center for Therapeutic Oncology Research. In addition to the American Type Culture Collection (ATCC), cell lines from cell banks all over the world can now be utilized by researchers [23]. Like syngeneic tumor models, cell line-derived xenograft (CDX) is also established by injecting mouse tumor cell lines expanded in vitro into mice. With the advantages of good reproducibility, rapid turn-around time, common availability, and cost-efficient generation, it is clear why CDX models are widely used in efficacy, pharmacokinetics (PK), and pharmacodynamics (PD) studies in drug discovery [24,25]. Among numerous examples, Suzawa et al. examined the antitumor activity of afatinib, an irreversible epidermal growth factor receptor (EGFR)-HER2 dual inhibitor, in several non-small cell lung cancer (NSCLC) cell lines, including A549, Calu-3, HCC827, NCI-H1299, NCI-H1781, NCI-H1975, NCIH1993, and NCI-H2170, to explore the correlation between the genetic alterations of cells and cell sensitivity to afatinib. The HER2-altered cell lines (H2170, Calu-3, and H1781) were found to be sensitive to afatinib, while the HER2- or EGFR-non-dependent NSCLC cells were insensitive, which implied that afatinib was a potential option for NSCLC patients with HER2 alterations [24].

However, CDX models have several significant shortcomings. First, the biological characteristics of cells in vitro for long-term passage may change. This leads to differences in tumorigenicity and the genomic and epigenetic properties of cell lines used in different laboratories. Second, a cell-line-derived xenograft has low heterogeneity compared to the original tumor. Third, CDXs are established in immune-deficient mice, thus limiting the application in the study of immunotherapy and immunomodulatory drugs [26,27,28,29,30].

## 5. Patient-Derived Xenografts (PDX)

Patient-derived xenografts (PDX) are established by subcutaneous or orthotopic implantation of surgical or biopsy tissue chunks into immune-deficient mice (Figure 1C) [31,32]. Tumor cells enriched from hydrothorax, ascites, and circulating tumor cells can also be used for PDX establishment [33,34,35]. After implantation, PDX requires a long latency period before tumor formation. The latency period can range from a couple of weeks to several months, depending on the type of cancer, the modeling method, and the growth characteristics of the original tumor [36]. PDX can be used for long-term preservation and pharmacodynamic studies after several cycles of serial in vivo transplantation [37]. At present, PDX has been successfully established in various cancers, including lung cancer [38], breast cancer [39,40], pancreatic cancer [41], colon and rectal cancer [42,43], gastric cancer [44], ovarian cancer [45], hepatocellular carcinoma [46], prostate cancer [47], brain tumors [48], melanoma [49], and head and neck tumors [50], etc. The success rates of modeling vary widely among cancer types from 10–100% as reported. A number of studies have proven that PDX can maintain the genetic profile, gene expression patterns, tissue histology, and drug response of the original tumor, as well as the molecular and cellular heterogeneity of the primary tumor [42,51]. In addition, PDX models also exhibit the possibility of predicting metastatic potential and drug response [52]. Currently, PDX models are widely considered a more physiologically relevant preclinical model than CDX and are widely used in the research of cancer mechanisms and the discovery of anti-tumor drugs [53,54,55]. Gao et al. established ~1000 patient-derived tumor xenograft models with multiple driver gene mutations to evaluate the population responses to 62 treatments from six cancer indications. Correlation between genotypes and drug response was identified from the research, showcasing the impressive progress in elucidating mechanisms of drug resistance and the profiling of therapeutic candidates [53].

Despite its significant advantages, PDX is less suited for large-scale drug screening and personalized precision medicine due to its long turn-around time and high cost. Usually, a 6–8-month period is required for tumor xenograft engraftment and efficacy studies [56,57,58]. Established in immune deficient mice, PDX models are not suitable for evaluating the efficacy of immunomodulators. In addition, one study showed that human stromal cells in PDX decreased gradually over passages, and were gradually replaced by mouse cells [59].

## 6. Conditionally Reprogrammed (CR) Cells

In addition to in vivo models, in vitro models, such as traditional cell lines, are also important tools for cancer research. However, primary tumor cells are difficult to proliferate in vitro, and the success rate of establishing cancer cell lines is as low as 1–10% [60,61]. Conditional reprogramming technology (CRT) effectively improves the success rate of primary cell culture and cell line establishment. One critical success factor of CRT is the use of feeder cells and ROCK inhibitors to help tumor cells and other epithelial cells proliferate continuously in vitro (Figure 1A) [62,63,64]. At present, multiple cancer cells have been established using CRT [64], including adenoid cystic carcinoma [65], breast [66], prostate [67,68], pancreatic [69], colorectal [70], lung [71,72], cervical [73], skin, kidney, ovarian cancers, and so on [64,74]. In addition, conditionally reprogrammed cells can be generated from PDX models and organoids [75,76,77] and passaged more than 200 times while preserving the heterogeneity of tumor cells [78,79]. As a suitable in vitro tool, CRT has been applied to drug development and precision medicine [65,68,80,81]. For example, Rei et al. established conditionally reprogrammed cells from patients with recurrent hormone receptor-positive (HR+)/human epidermal receptor 2-negative (HER2−) breast cancer to test heterogeneity and drug sensitivity. The mutation status and pathological features were preserved in CR cells, while the RNA expression was different from the primary tumor cells. The Rei group tested the responses of CR cells derived from an ER+/PgR+/HER2− liver metastasis to 224 drugs, and 66 compounds reduced cell viability, including SERD and CDK4/6 inhibitors. The original patient used SERD and CDK4/6 inhibitors after metastasectomy, and no recurrence was noted for 13 months, consistent with drug screening results on CR cells [80]. In another research project in prostate cancer, researchers generated CR cells from seven patients with diseases ranging from primary prostate cancer to advanced castration-resistant prostate cancer (CRPC). A high-throughput drug screening of 306 emerging and clinical cancer drugs led to the identification of navitoclax, an inhibitor of the Bcl-2 family, as the most effective drug against CRPC CR cells. In addition, taxanes, mepacrine, and retinoids were also proven potent in the screening [68].

Not only can CR cells be directly maintained in 2D culture, but they may also be grown into spheroids or organoids [67]. CR cells can also be inoculated into mice to obtain PDX models [62,63], which provides a new approach to establishing better performing models, including those for breast cancer and other cancers that have had low success rates in PDX.

One major limitation of CR cells or any in vitro model is that they are incapable of capturing the PK/PD properties of the compounds being evaluated. The concentration of the compounds being tested is typically constant, which is very different from the dynamic drug concentration in vivo due to the drug’s absorption, distribution, metabolism, and excretion (ADME) processes. Furthermore, in vitro cancer models are, in general, not suitable for the evaluation of prodrugs. For example, as a prodrug of 5-FU (5-fluorouracil), tegafur is slowly metabolized to 5-FU in the liver, thereby leading to lower toxicities than 5-FU [82]. Tegafur is clinically widely used in the treatment of numerous tumors [83,84], but its anti-tumor effect cannot be assessed in vitro directly in the absence of metabolizing enzymes.

## 7. Patient-Derived Organoid

Organoids are the organotypic structures established in vitro in 3D. The first adult stem cell-derived organoids were established from Lgr5-expressing mouse intestinal stem cells [85]. In recent years, organoid technology has been applied to the generation of organoids derived from cells isolated from patient cancer tissues and circulating tumor cells (Figure 1B) [86,87]. Organoids derived from patients can proliferate in vitro and be maintained in long-term cryopreservation. Besides the proliferation ability, organoids could also capture the heterogeneity of cancer.

Oded et al. established 56 organoid cultures from 32 patients, representing all main subtypes of ovarian cancer. The organoids recapitulated the hallmarks and tumor heterogeneity of ovarian cancer and demonstrated intra- and inter-patient heterogeneity. These organoids were also used for drug-screening assays to evaluate tumor subtype responses to chemotherapy [88]. Compared with PDX, organoids require a shorter experimental period and a lower cost, which can be used for medium–high throughput drug screening. At the same time, the establishment of patient-derived organoids allows the evaluation of anti-cancer drugs as an approach to identifying precision medicines. Pauli et al. collected 145 specimens of 18 different tumor types derived from 769 patients and subsequently established 56 patient-derived organoid cultures, which were used in high-throughput drug screening to discover effective treatment options [89]. It should be noted that several translational studies on organoids and precision medicine have delivered promising results [90,91,92,93,94]. For example, in the research of pancreatic cancer, patient-derived cancer organoids exhibited responses to chemotherapeutics, consistent with the efficacy observed in patients, and contributed to the evaluation of synchronous metastases. Based on the gene expression signatures of organoids, responses to chemotherapy and targeted drugs could be predicted [93]. In addition, Tiriac et al. collected 112 biopsy tissues of locally advanced rectal cancer (LARC) from patients enrolled in a phase III clinical trial of neoadjuvant chemoradiation (NACR) and established 96 patient-derived organoids (PDOs). Responses to radiation and chemotherapy on PDOs were consistent with patients’ clinical responses, with 84.43% accuracy, 78.01% sensitivity, and 91.97% specificity [94]. Organoids can also be engrafted, enabling in vivo model establishment and in vivo drug efficacy studies [88].

Although organoids are rapid and cost-effective, several shortcomings may limit their application in cancer research. The cell population of an organoid will gradually decrease during the culture process, and the heterogeneity is lower compared to PDX. In addition, it can only be administered in vitro and, hence, is not suitable for evaluating molecules in the context of systemic administration, metabolism, and distribution [61].

## 8. MiniPDX

In 2015, researchers proposed the concept of “next-generation functional diagnostics” [95] similar to next-generation sequencing (NGS), including 2D ex vivo drug toxicity testing, patient-derived xenografts (PDX), 3D organoids, and other technologies. The core purpose of these functional tests was to address the limitations of genome-based cancer therapeutic matching. MiniPDX, as an in vivo version of organoid technology, was developed to provide a more rapid and accurate drug efficacy test to guide personalized cancer treatments [96]. Fresh tumor specimens acquired from patients, including tumor tissues obtained from surgery, biopsy, puncture, pleural fluid, and ascites, are digested and purified to prepare cell suspensions. Then, the cell suspensions are filled into the MiniPDX capsules. After subcutaneous implantation, immunodeficient mouse-bearing capsules are treated for 7 days with candidate drugs or their combinations. Post-treatment, the implanted MiniPDX capsules are retrieved, and the relative luminance unit (RLU) is examined for tumor cell proliferation assessment. The calculated tumor cell growth inhibition (TCGI) (%) reflects the PDX response to treatment options in this model (Figure 1D).

The MiniPDX cancer drug sensitivity test has several favorable features, such as a short cycle time and high consistency, which are typically aligned with results from in vivo drug sensitivity tests and clinical treatment. MiniPDX drug sensitivity tests generate results in about 10 days, which can not only meet the requirements of preclinical research of antitumor drugs but also enable fast and accurate drug sensitivity results for patients in a clinical setting. In China, the MiniPDX drug sensitivity test model has benefited numerous tumor patients. For example, a patient with endometrial stromal sarcoma had lung metastasis two chemotherapy cycles after operation. The metastatic lesions were taken for a MiniPDX drug sensitivity test. It was found that the patient was sensitive to apatinib alone and apatinib combined with olaparib. Four months after changing the medication regimen, the patient’s lung metastasis began to subside [96]. In another example, the chemotherapy regimen supported by a MiniPDX drug sensitivity test significantly prolonged the survival time of patients with gallbladder cancer, with the median overall survival time increasing from 13.9 months to 18.6 months and the disease-free progression time increased from 12 months to 17.6 months [97]. Another study of metastatic duodenal adenocarcinoma combined MiniPDX with NGS to reveal the mutations of key genes while conducting rapid drug sensitivity tests to provide individualized treatment for patients [98]. In a study on patients with platinum-resistant ovarian cancer, the clinical treatment response rate guided by MiniPDX drug sensitivity detection can be as high as 75% [99]. To treat patients with non-small cell lung cancer, Chen et al. confirmed that the MiniPDX-guided treatment group showed a better OS and PFS than the conventional chemotherapy group [100]. Li and colleagues verified the drug response predicted by WES, proteomics, and phosphorproteomics on the MiniPDX model and established an entire workflow from the generation of large omics datasets to in vivo drug testing models of colorectal cancer [101]. MiniPDX drug sensitivity test results were largely consistent with clinical responses in more and more patients with different tumor types [102,103], which indicated that MiniPDX models have great potential for guiding personalized cancer therapy.

Although the MiniPDX drug sensitivity test model has made significant progress in clinical practice, some limitations exist, impacting the application and wide adoption of the MiniPDX model. For example, the MiniPDX drug sensitivity test model uses immunodeficient mice and thus cannot be used for the detection of antitumor immunotherapeutic drugs. In addition, animal rooms and cell culture rooms meeting specific regulations and qualifications are required to complete the MiniPDX test. Finally, due to the use of immunodeficient mice and the high operation and maintenance costs of animal experimental facilities, the cost of a MiniPDX drug sensitivity test is higher than that of a typical in vitro drug sensitivity test.

## 9. Conclusions and Perspective

There are various models for cancer research and anti-tumor drug discovery and development. The experimental methods and underlying mechanisms of these models are different (Figure 1), and each has its own advantages and disadvantages (Table 1). It is critical to select the most appropriate model in order to meet the needs of different research purposes. In vitro models, such as CRT and organoids, are relatively fast, cost-effective, and suitable for high-throughput screening. However, they are only suitable for evaluating certain types of drugs because of the absence of therapeutically relevant administration and PK/PD correlation.

Among the various in vivo models, syngeneic tumor models and GEMM are suitable for immune oncology studies, but gaps between the physiological characteristics of humans and mice limit their application. The CDX model is derived from human tumor cell lines, but it lacks heterogeneity due to the limited types of traditional cell lines. PDX has previously shown a high degree of translatability to patients, and heterogeneity after continuous passage can be maintained. However, due to its long turnaround time, PDX is more suitable for drug development than precision medicine. With the advantages of rapidness, cost-effectiveness, a wide application range, and a high success rate, MiniPDX is expected to become an important tool in precision medicine and anti-tumor drug research.

As anticancer drug discovery and development continue to evolve, the applications of the aforementioned models and their future refinement will play an important role in giving drug hunters and developers the evidence and confidence to advance molecules to clinical studies. This also applies for physicians making informed medication decisions. Even for the same research program and oncology target, data from multiple in vitro and in vivo models could corroborate, and help assess the potential of therapeutic intervention from different angles. The selection of cancer in vitro and in vivo models and the interpretation of data rely on a solid understanding of the study methods and the understanding of the advantages and limitations of these models.

## 10. Disclosure

Yuan Long and Bin Xie are employed by Shanghai LIDE Biotech Co., Ltd. Danyi Wen is the CEO and founder of Shanghai LIDE Biotech Co., Ltd., and she is an inventor of the patent application named “METHOD AND DEVICE FOR DRUG SCREENING”, filed by Shanghai LIDE Biotech Co., Ltd. The patent application inventors are Danyi Wen and Feifei Zhang. Hong C. Shen declares no conflicts of interest.

## Figures and Tables

**Figure 1 cells-11-03868-f001:**
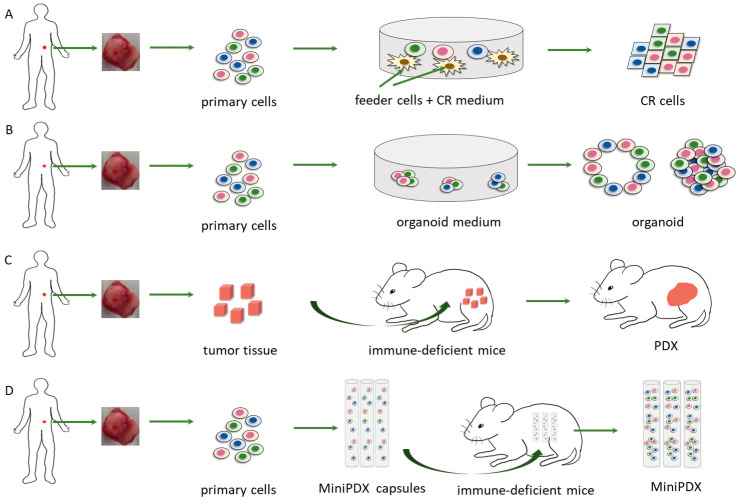
Schematic representation of CR cells, organoid, MiniPDX and PDX. For CR cells, organoids, and MiniPDX tumor tissues are diced and digested enzymatically to collect the primary suspension cells. (**A**). Tumor cells are co-cultured with feeder cells (irradiated or mitomycin C-treated mouse fibroblasts) and CR medium added ROCK inhibitor Y-27632. (**B**). Patient-derived organoids are established by embedment into a 3D matrix and specific medium to grow into 3D organotypic structures. (**C**). PDX is established by subcutaneous or orthotopic implantation of surgical or biopsy tissue chunks into immune-deficient mice. (**D**). Primary cells are filled into OncoVee^®^ capsules (LIDE Biotech Co., Ltd., Shanghai, China) and implanted subcutaneously via a small skin incision. The implanted capsules are removed after 7 days, and tumor cell proliferation is evaluated.

**Table 1 cells-11-03868-t001:** Characteristics of cancer models. The advantages and limitations of the cancer models mentioned are listed in Table 1, including features of biology, methodology, and application.

Feature	PDX	MiniPDX	Organoid	Conditional Reprogramming	CDX	GEMM	Syngeneic Model
Success rate of initiation	Low	High	Medium	Medium	High	High	High
Humanization	Yes	Yes	Yes	Yes	Yes	No	No
Initial sample source	Fresh clinical specimens	Fresh clinical specimens	Fresh clinical specimens	Fresh clinical specimens	Human cancer cell line	-	Mouse cancer cell line
Medium-dependency	No	No	Yes	Yes	No	No	No
Administration approaches	Systemically	Systemically	Through medium	Through medium	Systemi-cally	System-ically	Systemica-lly
Numbers of animals needed	High	Low	-	-	High	High	High
Facility requirements	High	High	Low	Low	High	Medium	Medium
Cost	High	Medium	Low	Low	Medium to high	High	Medium to high
High-throughput drug screening	No	No	Yes	Yes	No	No	No

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
