# Peer review of "Translation Potential and Challenges of In Vitro and Murine Models in Cancer Clinic"

_cells, 2022, doi:10.3390/cells11233868_

Round 1

Reviewer 1 Report

It is crucial to create and establish in vitro and in vivo models for the development of anticancer drugs and personalized cancer therapy.  Many such models were created in the past century, including syngeneic and xenograft tumor models derived from cell lines, tissue-based patient-derived xenograft models with or without humanized immune system, and genetic engineered animal models.  Sophisticated cell culture techniques provide excellent in vitro models including conditionally reprogrammed cells and organoid culture which greatly enhance our capability to evaluate cytotoxic effects of selected chemo drugs with patient derived primary tumor cells.  The authors made thorough reviews of those in vitro and in vivo models in terms of successful rate, cost of labor and animals, possibility of humanization, grade of heterogeneity withheld and clinical relevance, et al.  The authors highlighted the concept and application of the MiniPDX models in the drug discovery and development, especially the value of prediction in clinical application to guide personalized cancer therapy.  It is approved that the MiniPDX model is a rapid and efficient model combining the advantages of PDX models and 3D organoid models to achieve fast and cost-effective evaluation of therapeutic response in dynamic in vivo PD/PK environments. 

Author Response

Dear reviewer,

We appreciate the valuable comments from you.

The authors of this review are all experienced scientists in anti-cancer drug R&D. We summarized the characteristics of the in vitro and in vivo mouse models widely used in anti-cancer drug development basing on the research experience. We hope the review article could give the readers an insight into cancer models and could help the researchers to select the suitable models.

Reviewer 2 Report

Dear Ms. Harper Shang

Thank you for letting me review the paper of Long et al. The title of the review sounded interesting and promising. However, the manuscript itself does not reflect this as it is not a review but a minimalistic summary of a few models. The authors need to improve the quality of the manuscript, change it so the title and their message fit and mention a disclaimer.

The last is based on the fact that the authors use this review to promote their own service, the miniPDX. References to the use of this model are mostly self references or refer to facilities within their own logistic region.  

Even worse. Looking up the mentioned papers, none claims a competing interest while in my eyes there is a high conflict of interest here. This disclaimer is even not mentioned in this current manuscript. So, transparency and clear independent research is missing in the promotion of this product.

Author Response

Dear reviewer,

Thank you very much for the valuable comments from you and we appreciate the helpful suggestions on the disclosure of potential conflicts of interest.

As this is a review article, it is natural that we will summarize literature based on our own experience and domain knowledge. The self-referencing practice is common with most academic or industrial labs when contributing review articles. In addition, work from literature beyond our affiliation has been referenced (For example, Yao, Y., et al., Patient-Derived Organoids Predict Chemoradiation Responses of Locally Advanced Rectal Cancer. Cell Stem Cell, 2020. 26(1): p. 17-26 e6.). We regret to give you the perception that the review article is meant to promote the MiniPDX service of Shanghai LIDE Biotech Co.,LTD. To address this concern and enhance transparency, we will disclose that LIDE provides MiniPDX service to partners in the revised paper and also send the signed Disclosure of Potential Conflicts of Interest Form to the editor. Furthermore, it is our observation that the vast majority of publications from industry are to genuinely promote science instead of promote products/service. In this review, we summarized the characteristics of various models in the terms of including success rate, cost and heterogeneity, basing on the authors’ experience on anti-cancer drug research. We are keen on promoting science to benefit the scientific community via this review article, and we would appreciate your understanding.

Reviewer 3 Report

This is a very well-structured article, which gives a superficial review of the main animal models of cancer.

My main concern is about the scope of the review as far as animal species are concerned. The authors have only reviewed murine models, but there are more species in which useful models have been developed (e.g. rat, hamster, pig). Ideally, the authors should extend their review to include these species. But if this is not possible they should at least change the title of the publication, indicating that it refers exclusively to murine models.

Author Response

Dear reviewer,

Thank you very much for the valuable comments from you and we appreciate the helpful suggestions on the scope of animal species.

Although various in vivo models have been developed (e.g. rat, hamster, pig), the mouse tumor models are still the most widely used in the preclinical study of anti-cancer drugs. Since the efficacy data of mouse models could usually meet the requirements of preclinical studies and the study on mouse models are fast and cost effective, most researchers and pharmaceutical companies are using murine models instead of other animal species’ models on oncology drug R&D. Therefore, when we summarized the in vivo models we focused on the mouse models.

Thank you for pointing out that our title is not rigorous. We will change the title to "Applications of in vitro and murine models to enable discovery and development of anti-cancer drugs and clinical treatment" in the revised manuscript.

Round 2

Reviewer 2 Report

With the exception of adding a disclaimer to the manuscript, no changes have been made to improve the quality of the paper and therefore my first opinion that this a minimalistic summery of a few models is not changed. 

Author Response

Dear reviewer,

Thank you very much for your comments. We will continue to focus on anti-tumor research and strive to do better work in the future.